# Redox Isomerization of Allylic Alcohols Catalyzed by New Water-Soluble Rh(I)-*N*-Heterocyclic Carbene Complexes

**Csilla Enikő Czégéni** [1,*], **Marianna Fekete** [2], **Eszter Tóbiás** [2], **Antal Udvardy** [2],
**Henrietta Horváth** [1] **and Gábor Papp** [2] **and Ferenc Joó** [1,2]

[1] MTA-DE Redox and Homogeneous Catalytic Reaction Mechanisms Research Group, P.O. Box 400, H-4002 Debrecen, Hungary; henrietta.horvath@science.unideb.hu (H.H.); joo.ferenc@science.unideb.hu (F.J.)
[2] Department of Physical Chemistry, University of Debrecen, P.O. Box 400, H-4002 Debrecen, Hungary; feketemarijana@gmail.com (M.F.); tobias.eszter.de@gmail.com (E.T.); udvardya@unideb.hu (A.U.); papp.gabor@science.unideb.hu (G.P.)
* Correspondence: nagy.csilla@science.unideb.hu; Tel.: +36-52-512-900 (ext. 22588)

**Abstract:** New water-soluble, *N*-heterocyclic carbene (NHC) or mixed NHC/tertiary phosphine complexes [RhCl(cod)(sSIMes)], Na$_2$[Rh(bmim)(cod)(*m*tppts)], and [Rh(bmim)(cod)(pta)]BF$_4$ were synthetized and applied for the first time as catalysts in redox isomerization of allylic alcohols in aqueous media. [RhCl(cod)(sSIMes)] (with added sulfonated triphenylphosphine) and [Rh(bmim)(cod)(pta)]BF$_4$ catalyzed selectively the transformation of allylic alcohols to the corresponding ketones. The highest catalytic activity, TOF = 152 h$^{-1}$ (TOF = (mol reacted substrate) × (mol catalyst × time)$^{-1}$) was observed in redox isomerization of hept-1-en-3-ol ([S]/[cat] = 100). The catalysts were reused in the aqueous phase at least three times, with only modest loss of the catalytic activity and selectivity.

**Keywords:** allylic alcohols; redox isomerization; *N*-heterocyclic carbenes; rhodium; water soluble; homogeneous catalysis

## 1. Introduction

Catalytic isomerization of allylic alcohols is a remarkable way to obtain carbonyl compounds without classical oxidation/reduction steps, and is a 100% atom economic process of producing aldehydes and ketones (Scheme 1). The most efficient complexes for isomerization of allylic alcohols have been used in organic solvents [1–3]. However, water, as an environmentally friendly solvent for organic reactions, has received increasing attention. In aqueous oraqueous/biphasic systems the catalyst recycling is easier, and the process becomes more economical and environmentally friendly. Aqueous organometallic catalysis led to the development of a huge number of new and greener synthetic methodologies in organic synthesis. Lots of research has been done to develop efficient catalytic systems for the production of carbonyl compounds [4–8].

Rhodium complexes have been applied in numerous industrially significant reactions of hydrogenation [9], dehydrogenation of primary alcohols [10], hydroamination [11] and hydroformylation [12,13]. Bergens and co-workers reported an immobilized rhodium catalyst-organic framework for solvent-free isomerization of allylic alcohols [14]. A water soluble complex which forms during the reaction of [Rh(cod)(CH$_3$CN)$_2$]BF$_4$ (cod = $\eta^4$-cyclo-1,5-octadiene) and pta (1,3,5-triaza-7-phosphaadamantane) catalyzes the isomerization of codeine and morphine into hydrocodone and hydromorphone, which is an important transformation from a pharmaceutical point of view [15].

The most efficient water-soluble complexes contain water soluble tertiary phosphines, such as the sodium salts of mono-, di, and tri-*meta*-sulfonated triphenylphosphine (*m*tppms-Na, *m*tppds-Na$_2$, *m*tppts-Na$_3$, respectively) the sodium salt of sulfonated 1,3-bis-(diphenylphosphino)propane (dppp) [16], *para*-triphenylphosphine monophosphonate (*p*-tppmp) [17], and 1,3,5-triaza-7-phosphaadamantane (pta) [18]. Cationic Rh(I)-diphosphine catalysts were attached to proteins and used in asymmetric hydrogenations in aqueous solutions [19,20]. Rhodium(I)-*N*-heterocyclic carbene complexes have also been widely used in catalytic chemistry, however there have been relatively few reports on the application of Rh-NHC catalysts in water [6,11,16], despite the fact that water-soluble NHC complexes of other metal ions often serve as catalysts in aqueous solutions [11,21–26]. The first water-soluble Rh(I)-NHC complex, in which a functionalized *N*-heterocyclic ligand was responsible for the water solubility, was described in 1997 by Herrmann et al [27]. Kühn and co-workers reported the preparation and catalytic properties (in the hydrogenation of aromatic ketones) of sulfoalkyl-substituted azolium-derived water soluble Rh(I)-NHC complexes [28]. Klein Gebbink published the synthesis of the [NBu$_4$][RhCl(cod)(NHC)], and [PPh$_4$][RhCl(cod)(NHC)] (NHC = 1-(4-sulphonatobutyl)-3-(2,4,6-trimethylphenyl)imidazol-2-ylidene) [29]. Perrez-Torrente and co-workers synthesized and structurally characterized several water-soluble zwitterionic, carboxylate bridged *bis*-NHC rhodium complexes. The easy *N*-functionalization of NHC-ligands allowed stable attachment of Rh(I)-NHC-complexes to proteins, and the resulting biohybrid catalytsts were applied in aqueous systems for enantioselective hydrogenations by Klein Gebbink et al [30], and cascade metathesis/hydrogenation reactions by Okuda and co-workers [31].

**Scheme 1.** Synthetic ways of obtaining saturated carbonyls from allylic alcohols; (a) oxidation followed by hydrogenation, (b) hydrogenation followed by oxidation, (c) catalytic redox isomerization.

The use of rhodium complexes for the catalytic transformation of allylic alcohols to carbonyl compounds dates back more than four decades. Strohmeier and co-workers used 0.6 mol% [RhH(CO)(PPh$_3$)$_3$] to convert methallyl alcohol to isobutyraldehyde (3 h at 70 °C in trifluoroethanol, quantitative yield) [32]. In another interesting early work Alper and co-workers applied 2–4 mol% [Rh(CO)$_2$Cl]$_2$, together with phase transfer catalysts in aqueous–organic biphasic isomerizations [33]. Sasson and coworkers observed complete transformation of oct-1-en-3-ol to 3-octanone in benzene/water mixtures with 2% mol RhCl$_3$ [34]. Biphasic systems with water-soluble rhodium complexes of *meta*-trisulfonated triphenylphosphine (*m*tppts) derivatives were reported by de Bellefon [16,35]. Several other rhodium complexes were successfully applied in aqueous media

for the redox isomerization of allylic alcohols [18,36–40], however, no water-soluble Rh(I)-NHC- or Rh(I)-(NHC)-(phosphine) complexes have been reported for this purpose to date. Ru(II)-complex catalysts are more abundant in redox isomerization of allylic alcohols [41–43]. For example, NHC-ligated water-soluble Ru(II)-complexes were studied as catalysts in a reaction by Perís and co-workers [41]. In that work, water-solubility was due to sulfonate-functionalized NHC ligands. Mixed NHC/tertiary phosphine complexes of iridium(I) were also found to be active catalysts in the redox isomerization of allylic alcohols in aqueous systems [44].

In view of the scarcity of catalytic applications of Rh(I)-NHC complexes in redox isomerization of allylic alcohols in aqueous media, we synthesized new water-soluble complexes of this type with sulfonated *N*-heterocyclic carbene sSIMes (sSIMes = 1,3-bis(2,4,6-trimethyl-3-sodium-sulfonatophenyl)imidazolin-2-ylidene, as well as mixed-ligand Rh(I)-NHC-phosphine complexes with NHC = bmim = 1-butyl-3-methyl-imidazol-2-ylidene, and the water-soluble tertiary phosphine ligands *m*tppms-Na, and *m*tppts-Na$_3$. Synthesis of the catalysts, and their application in redox isomerization of allylic alcohols in aqueous systems, are described below. The new compounds were characterized with $^1$H, $^{13}$C, and $^{31}$P NMR spectroscopies, as well as with ESI mass spectrometry

## 2. Results and Discussion

An efficient synthetic strategy of Rh(I)-NHC complexes uses the easily available [RhCl(cod)]$_2$, [Rh(cod)(MeO)]$_2$ or [Rh(OH)(cod)]$_2$ dinuclear complexes as starting materials. Reaction of these compounds with imidazolium halides or tetrafluoroborates in the presence of bases often leads to clean formation of neutral or cationic mononuclear Rh(I)-NHC complexes [RhX(cod)(NHC)] and [Rh(cod)(NHC)]BF$_4$, both in organic and aqueous solvents. Further reaction of these mononuclear complexes with tertiary phosphines results in formation of mixed-ligand NHC/phosphine Rh(I)-complexes. We also followed this general strategy and used sulfonated NHC or/and water-soluble phosphine ligands to achieve water-solubility of the resulting complexes.

*2.1. Synthetic Procedure and Characterization of Rh(I)-N-Heterocyclic Complexes **1–5***

2.1.1. Synthesis of [RhCl(cod)(sSIMes)] (**1**) (Scheme 2).

The [RhCl(sSIMes)(cod)] (**1**) complex was prepared following the literature procedure, previously reported for water-insoluble Rh(I)-*N*-heterocyclic carbene complexes [45]. The synthesis involved reaction of [RhCl(cod)]$_2$ and the NHC ligand precursor, [sSIMesH]Cl, in the presence of excess K$_2$CO$_3$ in MeOH at reflux temperature (Scheme 2). The resulting complex is highly soluble in water and methanol, and insoluble in dichloromethane and other non-polar solvents. In the $^{13}$C{$^1$H} NMR spectrum recorded in MeOD, the resonance signal of the carbene carbon atom appears at 212.43 ppm as a doublet ($^1J_{Rh-C}$ = 48.22 Hz) which is in agreement with the values reported by Herrmann et al. for Rh-carbene carbon resonances (δ = 179.6–209.8 ppm, $^1J_{Rh-C}$ = 46.4 Hz–50.8 Hz) [45], and with the values obtained for water-insoluble analogues [46]. Unfortunately, despite all our attempts, **1** could not be isolated as a pure solid. $^1$H and $^{13}$C{$^1$H} NMR spectra showed decomposition during isolation, which is in accord with the reported instability of the [sSIMesH]Cl ligand in basic solvents [47], especially in solutions containing minute amounts of water. Conversely, methanolic solutions of **1**, (see Section 2.2) showed consistent catalytic activity, and were studied in aqueous solutions containing 13 % methanol.

Attempts were made to prepare the mixed ligand NHC-phosphine-Rh(I) complex (denoted **2** on Scheme 3) in the reaction of **1** and *m*tppms-Na. Nevertheless, all our efforts to isolate a uniform product failed. $^{31}$P{$^1$H} NMR of the resulting solution displayed a doublet at δ = 27.14 ppm with a coupling constant $^1J_{Rh-P}$ = 147.0 Hz, which unambiguously revealed the coordination of *m*tppms to Rh(I). Nevertheless, the addition of 1-equivalent of *m*tppms-Na to a solution of **1** largely increased the catalytic activity of **1** in the redox isomerization of allylic alcohols. Such solutions of **1**/*m*tppms-Na showed reproducible catalytic activity, and were used for the catalytic experiments (see below).

A probable cause of the failure in synthesis of the elusive compound **2** may be in the high steric demand of both the sSIMes and the *m*tppms ligands. Analogous NHC-phosphine-Ir(I) complexes with non-sulfonated NHC ligands, such as [Ir(bmim)(cod)(*m*tppms)] [44], and [Ir(cod)(emim)(*m*tppms)] (emim = 1-ethyl-3-methyl-imidazol-2-ylidene) [48] are known, however, those contain small size NHC ligands, compared to sSIMes. Another difference is that the mentioned Ir(I)-complexes showed only negligible solubility in aqueous solutions, since they were obtained as zwitterionic species, formed with loss of chloride and the sodium ion of the *m*tppms-Na ligand. In contrast, mixtures of **1** and 1-equivalent *m*tppms-Na did not yield a less soluble zwitterionic product, most probably due to the two $-SO_3^-$ substituents in the sSIMes ligand.

**Scheme 2.** Synthesis of **1** from [RhCl(cod)]$_2$ and [sSIMesH]Cl in MeOH.

**Scheme 3.** Attempted synthesis of [Rh(cod)(sSIMes)(*m*tppms)] in the reaction of **1** and monosulfonated triphenylphosphine (*m*tppms-Na).

2.1.2. Synthesis of [RhCl(bmim)(cod)] (**3**)

[RhCl(bmim)(cod)] (**3**) was previously prepared by Park et al. by the reaction between [Rh(MeO)(cod)]$_2$ and [bmimH]Cl, and the complex showed catalytic activity in hydrosilylation reactions [49]. We successfully used the less sensitive [Rh(OH)(cod)]$_2$ instead of [Rh(MeO)(cod)]$_2$, and its reaction with [bmimH]Cl yielded [RhCl(bmim)(cod)] (**3**), as depicted in Scheme 4.

Complex **3** is very poorly soluble in water, but the chloride on rhodium can be easily replaced by a water-soluble ligand. Due to this, water-soluble, phosphine-containing Rh(I)-NHC complexes were prepared by ligand exchange.

**Scheme 4.** Synthesis of complexes **3**–**5**.

### 2.1.3. Synthesis of Na$_2$[Rh(bmim)(cod)(*m*tppts)] (**4**)

Triply-sulfonated triphenylphosphine sodium-salt, *m*tppts-Na$_3$, is one of the most water-soluble tertiary phosphines (its solubility at room temperature is about 1400 g/L) [50]. It was expected that reaction of *m*tppts-Na$_3$ and [RhCl(bmim)(cod)] (**3**) would result in the formation of a highly water-soluble mixed ligand Rh(I)-NHC-phosphine complex, with low solubility in apolar organic solvents. Indeed, the reaction depicted in Scheme 4 yielded **4** as orange powder, highly soluble in water. In MeOD solution, the $^{31}$P{$^1$H} NMR spectrum of **4**, a doublet can be seen at δ = 27.31 ppm ($^1J_{Rh–P}$ = 160.4 Hz) (Figure S8), while the $^{13}$C{$^1$H} NMR signal of the carbene carbon atom appeared at δ = 174.62 ppm as a doublet of a doublet (Figure S7), because both carbon–rhodium(I) and carbon–phosphorus coupling occurred ($^1J_{Rh–C}$ = 49.2 Hz, $^2J_{C–P}$ = 15.4 Hz).

### 2.1.4. Synthesis of [Rh(bmim)(cod)(pta)]BF$_4$ (**5**)

Starting from [Rh(OH)(cod)]$_2$, another water-soluble *N*-heterocyclic carbene complex, **5**, was obtained in reaction with pta as a water-soluble phosphine ligand (Scheme 4). Complex **5** was prepared in two ways.

According to Method A, [Rh(OH)(cod)]$_2$ was refluxed with 1-butyl-3-methylimidazolium-tetrafluoroborate ([bmimH]BF$_4$ and pta. In this case, [bmimH]BF$_4$ protonates the [Rh]–OH group, followed by water dissociation and simultaneous carbene coordination. NMR spectroscopic measurements showed that the resulting intermediate [Rh*S*(bmim)(cod)]BF$_4$ (*S* = solvent) was not sufficiently pure. However,

the addition of pta led to formation of the stable [Rh(bmim)(cod)(pta)]BF$_4$ (**5**), which was isolated as an orange powder.

Method B of the synthesis of **5** was based on chloride removal from **3**. [RhCl(bmim)(cod)] was stirred with NaBF$_4$ in MeOH at room temperature. The resulting NaCl was removed by filtration, then 1-equivalent of pta was added to the solution (Scheme 4). The coordination of the phosphine ligand to rhodium(I) was displayed by $^{31}$P{$^1$H} NMR. The spectrum recorded in MeOD showed a doublet (Figure S11) at δ = −54.90 ppm ($^1J_{Rh-P}$ = 125.6 Hz), while the $^{13}$C{$^1$H} NMR resonance of the carbene carbon appeared as a doublet of a doublet, at δ = 176.71 ppm ($^1J_{Rh-C}$ = 48.5 Hz, $^3J_{C-P}$ = 18.5 Hz). Formation of **5** was unambiguously identified by the appearance of the [Rh(bmim)(cod)(pta)]$^+$ (506.1915 Da) molecular ion peak in the ESI-TOF MS spectrum, with an exact match of the experimentally determined and calculated isotope distributions. (Figure S15).

### 2.1.5. Single-crystal X-ray diffraction analysis of [Rh(bmim)(η$^4$-cod)(pta)]BF$_4$ (**5**)

Single crystals of **5** were obtained by crystallization from MeOD at −18 °C. The compound crystallized in the monoclinic *P*2$_1$/*n* (Nu. 14) space group. The asymmetric unit contains one cationic complex ion (Figure 1) and one disordered BF$_4^-$. (with 32% and 68% occupancies). The structures of [RhCl(cod)(bmim)] [49,51] and [Rh(cod)(SCN)(bmim)] [52] have been previously determined. In these phosphine-free complexes, the Rh–C$_{carbene}$ bond distances are 2.019(2) Å [51], 2.023(6) Å [49], 2.046(2) Å [52], while in **5** the same distance is 2.025(4) Å, i.e., it is in accordance with other carbene rhodium complexes. At the same time, the Rh–P distance in **5** is 2.2668(17) Å, which is shorter than the same bond distance in [Rh(cod)(*i*PrMeIm) (PPh$_3$)]BF$_4$ × CH$_2$Cl$_2$, 2.332 Å (*i*PrMeIm = 1-isopropyl-3-methylimidazol-2-ylidene), GOLMAX [53], and in [Rh(cod)(BnMesIm)(PPh$_3$)]BF$_4$, 2.363 Å (BnMesIm = 1-benzyl-3-mesitylimidazol-2-ylidene) GOHBIQ [54]. The shortening of the Rh–P bond is the consequence of the stronger coordination of pta to the [Rh(cod)(NHC)]$^+$ compared to the bonding of PPh$_3$. The C$_{carbene}$–Rh1–P1 bond angle is small, 88.44(11)°, which reflects the smaller space requirement of pta than that of PPh$_3$ (the same angles in the [Rh(cod)(NHC)PPh$_3$] complexes are 96.56° (*i*PrMeIm) and 91.89° (BnMesIm)). Since both pta and bmim are rigid ligands, significant differences were not expected (and not found) in the major bond lengths and angles (Table S1 and Table S2).

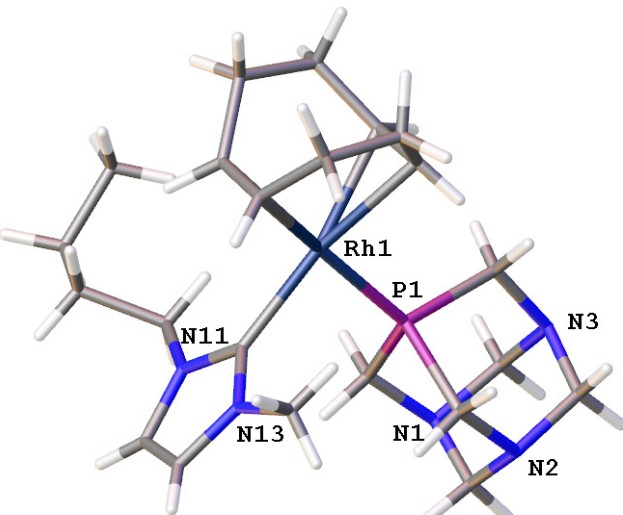

**Figure 1.** Capped stick representation of **5**. (Disordered BF$_4^-$ is omitted for clarity.).

## 2.2. Redox isomerization of allylic alcohols with water soluble Rh(I)-NHC catalysts

The new water-soluble Rh(I)-NHC and Rh(I)-NHC-phosphine complexes were studied as catalysts for redox isomerization of allylic alcohols in aqueous media (Scheme 5). In most cases, oct-1-en-3-ol was used to optimize the conditions.

**Scheme 5.** General scheme of redox isomerization of allylic alcohols.

The isomerization reactions were investigated using [RhCl(cod)(sSIMes)] (**1**), and the catalyst prepared in situ from **1** and 1-equivalent *m*tppms-Na. The effects of various added phosphines, *m*tppms-Na, *m*tppts-Na$_3$, pta, and PPh$_3$ were also studied. The results are shown in Table 1.

**Table 1.** Redox isomerization/hydrogenation of oct-1-en-3-ol with [RhCl(sSIMes)(cod)] (**1**).

| Entry | Catalyst | [S]/[Rh] | Added PR$_3$ Phosphine | [P]/[Rh] | T (°C) | Octan-3-on (%) [a] | Octan-3-ol (%) [a] |
|---|---|---|---|---|---|---|---|
| 1 | **1** | 50 | - | - | 55 | 15 | 0 |
| 2 | **1** + H$_2$ (1 bar) | 50 | - | - | 55 | 18 | 2 |
| 3 | **1** | 50 | *m*tppms | 1 | 55 | 43 | 0 |
| 4 | **1** | 50 | *m*tppms | 2 | 55 | 24 | 0 |
| 5 | **1** | 50 | *m*tppms | 3 | 55 | 13 | 0 |
| 6 | **1** | 50 | *m*tppms | 1 | 80 | 98 | 0 |
| 7 [b] | **1** | 100 | *m*tppms | 1 | 80 | 58 | 0 |
| 8 [b] | **1** | 100 | *m*tppts | 1 | 80 | 44 | 0 |
| 9 [b] | **1** | 100 | pta | 1 | 80 | 85 | 0 |
| 10 [b] | **1** | 100 | PPh$_3$ | 1 | 80 | 98 | 0 |
| 11 | - | 50 | *m*tppms | - | 80 | 0 | 0 |
| 12 | [RhCl(cod)]$_2$ | 50 | - | - | 80 | 3 | 0 |

Conditions: $2.89 \times 10^{-1}$ M oct-1-en-3-ol, $5.79 \times 10^{-3}$ M [Rh], [a] Yields determined by gas chromatography. [b] $2.89 \times 10^{-3}$ M [Rh], 3 mL H$_2$O + 450 μL MeOH, 2 h, argon atmosphere (except entry 2).

With the use of Rh(I)-NHC catalyst (**1**), the reaction proved selective for the isomerized product, but the conversion was low (entry 1). Running the reaction in an H$_2$ atmosphere of 1 bar pressure did not increase the total conversion significantly, however, octan-3-ol also appeared among the products (entry 2). Consequently, all further experiments were conducted under an argon atmosphere. Under such conditions, addition of 1-eq. *m*tppms-Na to **1**, increased the activity by about a factor of three (entries 3 vs 1). Concerning the effect of added tertiary phosphines (entries 7–10), the highest conversion was achieved with addition of PPh$_3$, while the lowest catalytic activity was observed in the presence of *m*tppts. An increase of the [PR$_3$]/[Rh] molar ratio above 1 resulted in strong inhibition of redox isomerization of oct-1-en-3-ol (entries 3–5). In the control experiments (entries 11, 12), *m*tppms did not show any isomerization activity, while the use of [RhCl(cod)]$_2$ led to a mere 3% conversion, in contrast to the value of 98% achieved with **1**/*m*tppms-Na under identical conditions (entry 6). Under slightly different conditions (aqueous phosphate buffer, 1 bar H$_2$, pH 7.0, 80 °C), **4** was found to have a preference for hydrogenation (58% yield in 1 h) over redox isomerization (48% yield), and was not further scrutinized in detail.

Figure 2 shows the progress of the redox isomerization of oct-1-en-3-ol with **1**/*m*tppms-Na as the catalyst. The reaction proceeded smoothly according to a saturation curve, was selective to formation of octan-3-one, and led to a conversion of 98% in one hour at 80 °C.

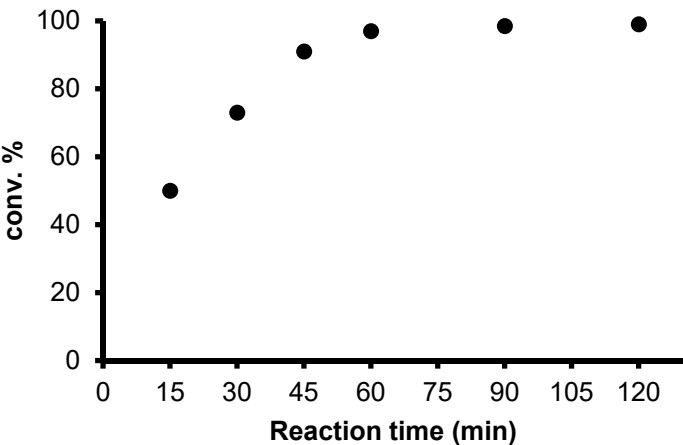

**Figure 2.** Time course of the redox isomerization of 1-octen-3-ol catalyzed by **1**/*m*tppms-Na. Conditions: $2.89 \times 10^{-1}$ M oct-1-en-3-ol, $5.79 \times 10^{-3}$ M **1**, $5.79 \times 10^{-3}$ M *m*tppms-Na, 3 mL $H_2O$ + 450 µL MeOH, 2 h, 80 °C, argon atmosphere.

Study of the effect of temperature on the rate of isomerization revealed that the reaction proceeded slowly at low temperatures (below 40 °C, Figure 3). In contrast, above 55 °C an increase of the reaction rate was observed, so much so, that at 80 °C, 99% conversion was achieved in 2 hours. Therefore, further measurements were carried out at 80 °C. As seen on Figure 3, the temperature dependence of the reaction rate did not follow the Arrhenius relation. However, it should be mentioned, that this relation is valid only to the temperature dependence of rate coefficient(s) of known kinetic equations. Data at high conversions are not suitable to represent the reaction rate, furthermore, in aqueous–organic biphasic systems, transport of the substrate to the aqueous phase may have a smaller temperature dependence than the catalytic reaction itself. Altogether, these causes may lead to deviations from the expected exponential rate increase.

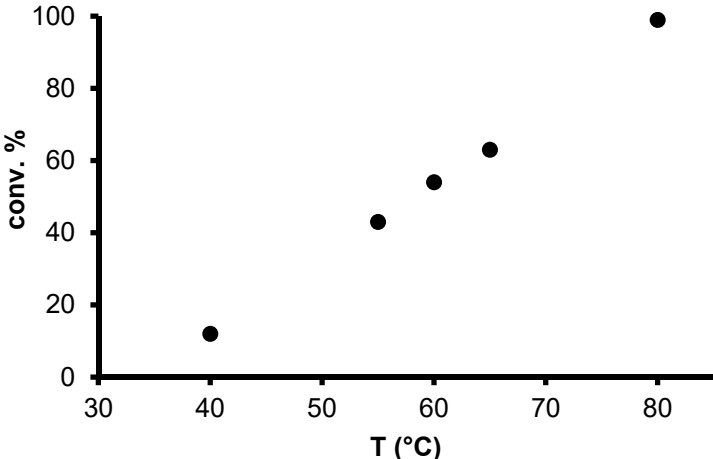

**Figure 3.** Temperature effect on redox isomerization of oct-1-en-3-ol catalyzed by **1**/*m*tppms-Na. Conditions: $2.89 \times 10^{-1}$ M oct-1-en-3-ol, $5.79 \times 10^{-3}$ M **1**, $5.79 \times 10^{-3}$ M mtppms-Na, 3 mL $H_2O$ + 450 µL MeOH, 2 h, argon atmosphere.

From an environmental and economic point of view, the amount of the applied catalyst cannot be neglected. Increasing the amount of the catalyst in the reaction mixture, resulted in a linear increase of the conversion (Figure 4). Beyond practical considerations, this rate dependence suggests a 1st order kinetics with regard to the catalyst concentration. Nevertheless, it has to be born in mind, that the detailed study of the reaction kinetics was not considered as a part of the present study, and the data

shown on Figures 2–4 rather serve the optimization of a synthetic procedure than the scrutiny of the underlying molecular events.

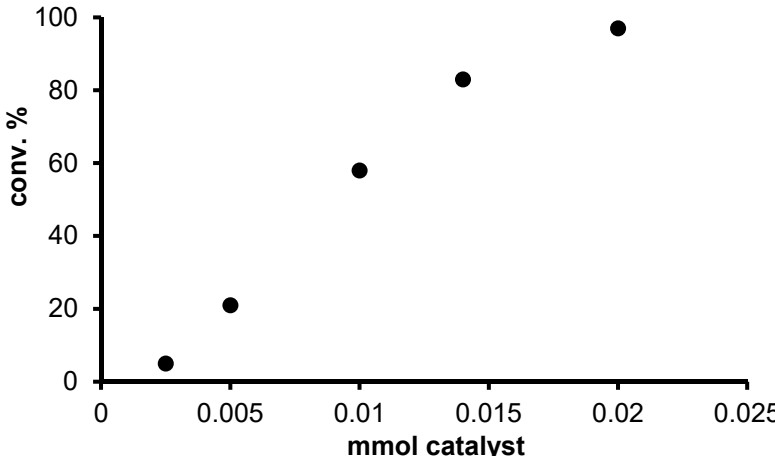

**Figure 4.** Effect of catalyst loading on the redox isomerisation of oct-1-en-3-ol catalyzed by **1**/*m*tppms-Na. Conditions: $2.89 \times 10^{-1}$ M oct-1-en-3-ol, $2.89 \times 10^{-3}$ M **1**, $2.89 \times 10^{-3}$ M *m*tppms-Na, 3 mL $H_2O$ + 450 μL MeOH, 1 h, 80 °C, argon atmosphere.

Due to the in situ preparation of the catalyst solution, most of the reaction mixtures contained 13.0% V/V methanol. In such a solution the conversion increased to 60% from 40%, determined in water alone as solvent. The increased conversion was most probably due to the increased solubility of oct-1-en-3-ol in the mixed solvent. Nevertheless, these data show, that an aqueous solution of **2** is suitable as a catalyst for the redox isomerization of oct-1-en-3-ol, as well as in the absence of any co-solvent. In such cases, however, an aqueous–organic biphasic reaction mixture is obtained, in which the organic phase is formed by the substrate itself.

One important reason for application of water-soluble catalysts in organic synthesis is the possible recirculation of the catalyst solution. Provided the catalyst is insoluble in the organic phase made up by appropriate solvents and/or the substrates/products, recovery of the catalyst can be achieved by liquid–liquid phase separation. In favorable cases, not only can the catalyst be recycled, but the product can be collected practically free of catalyst residues (metal contamination), which is a very important aspect for the pharmaceutical industry. Complexes containing highly ionic ligands, such as sulfonated water-soluble phosphines and/or *N*-heterocyclic carbenes, usually have very low solubility in organic solvents and thus do not pollute the product.

We checked the recovery and reuse of the catalyst **1**/*m*tppms-Na in redox isomerization of oct-1-en-3-ol (Table 2). After each catalytic cycle, the reaction mixture was cooled to room temperature, and was extracted with hexane. After phase separation, the catalyst-containing aqueous phase was used again to catalyze the reaction of a new batch of oct-1-en-3-ol. In the first recycle, the catalytic activity of complex **2** dropped approximately 20%, however, in the third recycle it did not decrease significantly.

**Table 2.** Recycling of the catalyst (**1**/*m*tppms-Na) in redox isomerization of oct-1-en-3-ol.

| Entry | Run | Octan-3-on (%) [a] | TOF ($h^{-1}$) |
|:---:|:---:|:---:|:---:|
| 1 | 1 | 82 | 27 |
| 2 | 2 | 68 | 23 |
| 3 | 3 | 64 | 21 |

Conditions: $2.89 \times 10^{-1}$ M oct-1-en-3-ol, $5.79 \times 10^{-3}$ M **1**, $5.79 \times 10^{-3}$ M *m*tppms-Na, 3 mL $H_2O$ + 450 μL MeOH, 1.5 h, 70 °C, argon atmosphere. [a] GC yields.

We undertook a study of the isomerization of various allylic alcohols with catalysts **1**/*m*tppms-Na and **5,** and the summarized results are presented in Table 3. The water-insoluble (or only slightly soluble) allylic alcohols were isomerized with high turnover frequencies, however the water-soluble substrates like prop-1-en-3-ol and but-1-en-3-ol showed smaller conversions. The obtained TOF values calculated from conversions determined at 30 min reaction times, were between 13–152 h$^{-1}$.

**Table 3.** Redox isomerization of allylic alcohols with Rh(I)-NHC-phosphine catalysts in water.

| Substrate | 1/*m*tppms-Na | | 5 [a] | |
|---|---|---|---|---|
| | Conversion (%) | TOF(h$^{-1}$) | Conversion (%) | TOF(h$^{-1}$) |
| oct-1-en-3-ol | 73 [b] | 146 | 18 | 13 |
| hept-1-en-3-ol | 76 | 152 | 53 | 38 |
| hex-1-en-3-ol | 75 | 150 | 83 | 59 |
| pent-1-en-3-ol | 70 [c] | 140 | 94 | 67 |
| but-1-en-3-ol | 45 [c] | 90 | 75 | 51 |
| prop-1-en-3-ol | 44 [c] | 88 | 73 | 52 |

Conditions: $3.10 \times 10^{-1}$ M substrate, $3.10 \times 10^{-3}$ M **1**, $3.10 \times 10^{-3}$ M *m*tppms-Na; 225 µL MeOH + 3 mL H$_2$O; argon atmosphere; [a] pH = 7.0 (0.1 M phosphate buffer) $4.66 \times 10^{-3}$ M **5**, 1 h, 80 °C; [b] $2.89 \times 10^{-1}$ M substrate, $5.79 \times 10^{-3}$ M **1**, $5.79 \times 10^{-3}$ M *m*tppms-Na; 3 mL H$_2$O + 450 µL MeOH, 30 min, 80 °C; conversions determined by gas chromatography; [c] $3.33 \times 10^{-1}$ M substrate, $3.33 \times 10^{-3}$ M [Rh], 3 mL H$_2$O, conversions determined by $^1$H NMR spectroscopy.

In aqueous organometallic catalysis, one of the most important reaction parameters is the solution pH, especially in cases when the reactions proceed with formation or consumption of H$^+$ (a typical example is the heterolytic splitting of H$_2$) [55]. For that reason, we investigated the effect of the pH of the reaction mixture on the isomerization of 2-methylprop-1-en-3-ol, using **5** as the catalyst. As can be seen on Figure 5, the conversion of the substrate allylic alcohol varied according to a maximum curve, with the highest conversions around pH 7.5.

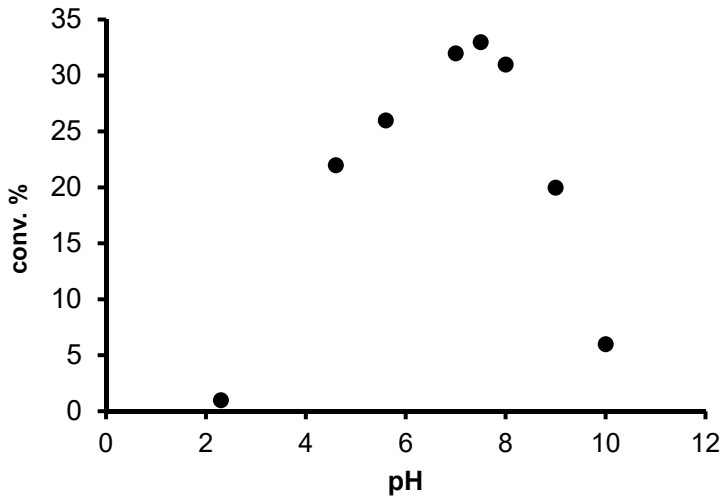

**Figure 5.** Conversion of the redox isomerization of 2-methylprop-1-en-3-ol, catalyzed by **5**, as a function of the pH. Conditions: $3.33 \times 10^{-1}$ M oct-1-en-3-ol, $4.66 \times 10^{-3}$ M **5**, 3 mL 0.1 M phosphate buffer, 1 h, 80 °C, argon atmosphere.

Similar maxima in the conversion vs. pH functions were observed in redox isomerization of oct-1-ene-3-ol catalyzed by the Ru(II)-arene complexes, [RuCl$_2$(bmim)($\eta^6$-*p*-cymene)] (*p*-cymene = 4-isopropyltoluene) [56], and [RuCl(Cp)(Me-pta)$_2$](OSO$_2$CF$_3$)$_2$ (Cp = $\eta^5$-cyclopentadienyl; Me-pta = *N*-methyl-pta) [57].

As discussed earlier, the rate of the redox isomerization of allylic alcohols was a linear function of the catalyst concentration, and the reaction was inhibited by an excess of *m*tppms-Na. Furthermore, the catalysts [Rh(bmim)(cod)(pta)]BF$_4$ (**5**), and with lesser activity, [RhCl(cod)(sSIMes)] (**1**) did not

need the presence of $H_2$ gas for catalysis of the reaction. Finally, the rate of redox isomerization of 2-methylprop-1-en-3-ol, catalyzed by **5,** showed a sharp maximum as the function of pH. All these findings are consistent with the $\eta^3$-oxo-allyl mechanism of such reactions (Figure 6).

**Figure 6.** The suggested $\eta^3$-oxo-allyl mechanism of the redox isomerization of allylic alcohols in aqueous media catalyzed by the Rh(I)-complexes **1**, **1**/*m*tppms-Na, and **5** studied in this work.

According to the suggested mechanism, in the first step of the catalytic cycle, a deprotonated allylic alcohol coordinates to the metal complex, both with its olefinic bond, and the oxygen donor atom. This step is facilitated in more basic solutions but is retarded by an excess of the phosphine ligand, provided that the required free coordination site on Rh(I) is created by phosphine dissociation. Subsequent β-hydride elimination results in formation of a hydrido-metal-enone intermediate which rearranges to a π-oxo-allyl complex. Protonation (from the solvent) of this intermediate leads to formation of the free enol, which then rearranges to the more stable carbonyl product. This last step is more facile in acidic solutions, and the proton production and consumption in the first and lasts steps of the catalytic cycle, respectively, may lead to the observed maximum of the rate as the function of pH. This description of the reaction mechanism agrees with the limited experimental findings, however, for establishing a sound mechanistic suggestion more detailed investigations are required.

## 3. Materials and Methods

### 3.1. Materials

All materials and reagents were obtained commercially and used as received from Sigma-Aldrich (St. Louis, MO, USA); VWR International (West Chester, PA, USA); and Molar Chemicals Kft. (Halásztelek, Hungary).

Merck Kieselgel 60 F254 plates (Merck, Darmstadt, Germany) were used for analytical thin-layer chromatography (TLC), the chromatograms were visualised by UV fluorescence at 254 nm. For column chromatography, silica gel (70–230 mesh, 63–200 µm (Sigma-Aldrich, St. Louis, MO, USA) was used. Linde Magyarország Zrt. (Répcelak, Hungary) supplied the Ar, $H_2$, and $N_2$ gases.

The water-soluble phosphine ligands pta [58], *m*tppts-Na$_3$ [59], and *m*tppms-Na [60], the water soluble [sSIMesH]Cl [25], and the metal precursors [RhCl(cod)]$_2$ [61], [Rh(OH)(cod)]$_2$ [62] were prepared according to literature methods.

## 3.2. General Methods

$^1$H and $^{13}$C{$^1$H}, $^{31}$P{$^1$H} NMR spectra were recorded on Bruker Avance 360 MHz and Bruker DRX 400 NMR spectrometers (Bruker, Billerica, MA, USA) and were referenced to residual solvent peaks and to 85% phosphoric acid. A Bruker maXis II MicroTOF-Q type Qq-TOF-MS instrument (Bruker Daltonik, Bremen, Germany) was used to obtain high-resolution electrospray ionization mass spectra (HR ESI-MS) in positive ion mode, and controlled by Compass Data Analysis 4.4 software from Bruker.

The X-ray intensity data were measured on an Enraf Nonius TurboCAD4 diffractometer. The structure was solved by ShelXT [63], refined with ShelXL [64], using Olex$^2$ [65] and WinGX [66] graphical interfaces. Publication materials were prepared by Mercury [67] and Platon [68]. The crystallographic data (including structure factors) for **5** were deposited in the Cambridge Crystallographic Data Centre (CCDC) with the No. CCDC 2023158.

Gas chromatographic analyses were performed with the use of an Agilent Technologies 7890 A instrument (HP-5, 0.25 μm × 30 m × 0.32 mm, FID 300 °C, (Agilent Technologies, Santa Clara, CA, USA); carrier gas: nitrogen 1.9 mL/min). The products were identified by comparison of their retention time with well-known standard compounds.

### 3.2.1. Preparation of [RhC(cod)(sSIMes)] (**1**)

In 7 mL MeOH 0.122 g (0.22 mmol) [sSIMesH]Cl was dissolved in a Schlenk tube equipped with a reflux condenser. To this solution at room temperature, 0.055 g (0.11 mmol) [RhCl(cod)]$_2$ and 0.307 g (2.22 mmol) K$_2$CO$_3$ were added in one portion. The resulting yellow solution was refluxed for four hours under argon atmosphere. The reaction mixture was cooled to room temperature and filtered through a Celite Hyflo Supercel pad, and the solvent was evaporated under vacuum. The product, a hygroscopic sticky solid was dissolved in 5 mL dry MeOH. Aliquots of this stock solution were used for further experiments.

$^1$H-NMR (360 MHz, MeOD) δ/ppm: 7.42 (s, 2H, C$H_{Ar}$) 4.74–4.49 (br, 2H, -C$H_2$-, cod), 4.17 (s, 4H, N-C$H_2$-C$H_2$-N), 3.69–3.60 (br, 2H, -C$H_2$-, cod), 2.96 (s, 12H, CH$_3$), 2.63 (s, 6H, CH$_3$), 2.24–2.15 (m, 4H, =CH-CH$_2$-, cod), 1.80–1.78 (m, 4H, =CH-CH$_2$-);

$^{13}$C-NMR (90 MHz, MeOD) δ/ppm: 142.8; 139.8; 137.8; 137.2; 135.4; 132.7; 97.8 (d, $^1J_{Rh–C}$ = 7.9 Hz); 69.1 (d, $^1J_{Rh–C}$ = 13.2 Hz); 51.7; 32.0; 27.2; 22.5; 17.7; 16.7.

MS(ESI), positive mode, in MeOH, *m/z* for **1**, [M-Cl]$^+$ (C$_{29}$H$_{36}$N$_2$Na$_2$O$_6$RhS$_2$), Calculated: 721.0860, Found: 721.0859.

### 3.2.2. Synthesis of [RhCl(bmim)(cod)] (**3**)

In an argon-filled Schlenk tube in 20 mL distilled CH$_2$Cl$_2$ 0.327 g (0.717 mmol) [Rh(OH)(cod)]$_2$ was dissolved, and then 5 mL CH$_2$Cl$_2$ 0.25 g (1.434 mmol) [bmimH]Cl was added. The solution was stirred for 6 h at reflux temperature, and then the solvent was removed under vacuum yielding a dark yellow sticky residue. This residue was cooled in liquid N$_2$, and triturated several times with small portions of cold diethyl ether, yielding **3** as a light yellow powder. Yield: 0.413 g (1.08 mmol), 75%.

$^1$H NMR (360 MHz, CD$_2$Cl$_2$) δ/ppm: 6.90 (s, 2H, -N-C$H$=C$H$-N-) δ: 4.98–4.89 (m, 2H, N-C$H_2$CH$_2$CH$_2$CH$_3$), 4.56–4.42 (m, 2H, -C$H_2$-, cod), 4.10 (s, 3H, N-C$H_3$), 3.40–3.26 (m, 2H, -C$H_2$-, cod), 2.54–2.33 (m, 4H, =CH-CH$_2$-, cod), 2.07–1.79 (m, 4H, =CH-CH$_2$-, cod + 2H, N-CH$_2$C$H_2$CH$_2$CH$_3$), 1.5 (m, 2H. N-CH$_2$CH$_2$C$H_2$CH$_3$) 1.07 (t, 3H, N-CH$_2$CH$_2$CH$_2$C$H_3$);

$^{13}$C{$^1$H} NMR (90 MHz, CD$_2$Cl$_2$) δ/ppm: 182.11 (d, $^1J_{Rh–C}$ = 45.79 Hz, NCN), 121.95 (-NCH=CHN-), 120.14 (-N-CH=CH-N-), 97.83 (d, $^1J_{Rh–C}$ = 7.4 Hz, =CH-CH$_2$-, cod), 97.72 (d, $^1J_{Rh–C}$ = 7.4 Hz, =CH-CH$_2$-, cod), 68.08 (d, $^1J_{Rh–C}$ = 14.8 Hz, -CH=CH-, cod) 67.28 (d, $^1J_{Rh–C}$ = 14.82 Hz, -CH=CH-, cod), 50.39 (-N-CH$_2$CH$_2$CH$_2$CH$_3$), 37.55 (CH$_3$-N-), 33.29, 32.99 (-CH$_2$-, cod), 33.52 (-N-CH$_2$CH$_2$CH$_2$CH$_3$), 29.18, 28.52 (-CH$_2$-, cod), 20.05 (-N-CH$_2$CH$_2$CH$_2$CH$_3$), 13.60 (-N-CH$_2$CH$_2$CH$_2$CH$_3$);

MS(ESI), positive mode, in MeOH, *m/z* for **3**, [M]$^+$ (C$_{16}$H$_{26}$N$_2$Rh), Calculated: 349.1146 Found: 349.1145.

### 3.2.3. Synthesis of Na$_2$[Rh(bmim)(cod)(*m*tppts)] (**4**)

In an argon-filled Schlenk tube, 0.200 g (0.520 mmol) [RhCl(bmim)(cod)] (**3**) was dissolved in 20 mL acetone. To the resulting clear yellow solution 0.323 g (0.520 mmol) *m*tppts-Na$_3$ was added followed by 3.5 mL deoxygenated water, upon which the colour of the solution became dark yellow. After 5 min stirring the solvent was evaporated in vacuum, and the resulting sticky residue, cooled in liquid N$_2$, was triturated with small portions of cold diethyl ether, yielding **4** as a yellow solid. Yield: 0.212 g (0.210 mmol), 40%.

$^1$H NMR (400 MHz, MeOD) δ/ppm: 8.53–7.20 (m, 12H, Ar–C*H*-, *m*tppts), 7.00 (s, 2H, -N-C*H*=C*H*-N-), 4.81–4.44 (m, 4H, -C*H$_2$*-, cod), 4.27 (m, 2H, N-C*H$_2$*CH$_2$CH$_2$CH$_3$), 3.78 (s, 3H, N-C*H$_3$*), 2.61–2.27 (m, 8H. =CH-C*H$_2$*-, cod), 1.64–1.59 (m, 2H, N-CH$_2$C*H$_2$*CH$_2$CH$_3$), 1.46–1.43 (m, 2H, N-CH$_2$CH$_2$C*H$_2$*CH$_3$), 0.99 (t, 3H, N-CH$_2$CH$_2$CH$_2$C*H$_3$*);

$^{13}$C{$^1$H} NMR (100 MHz, MeOD) δ/ppm: 174.62 (dd, $^1J_{Rh-C}$ = 49.2 Hz, $^2J_{C-P}$ = 15.5 Hz. NCN), 146.08–128.89 (m, Ar-C*P*), 124.54, 121.78 (-N-CH=CH-N-), 98.71 (dd, $^1J_{Rh-C}$ = 9.2 Hz, $^3J_{C-P}$ = 9.9 Hz =CH-CH$_2$-, =CH-CH$_2$-, cod), 97.5 (dd, $^1J_{Rh-C}$ = 7.9 Hz, $^3J_{C-P}$ = 9.9 Hz, =CH-CH$_2$-, cod), 94.55 (dd, $^1J_{Rh-C}$ = 13.9 Hz, $^3J_{C-P}$ = 7.3 Hz, -CH=CH-, cod), 50.52 (N-*C*H$_2$CH$_2$CH$_2$CH$_3$), 36.99 (*C*H$_3$-N-), 33.33, 31.99(-*C*H$_2$-, cod), 30.38 (-N-CH$_2$*C*H$_2$CH$_2$CH$_3$), 29.77, 29.25 (-*C*H$_2$-, cod), 19.76 (-N-CH$_2$CH$_2$*C*H$_2$CH$_3$), 12.97 (N-CH$_2$CH$_2$CH$_2$*C*H$_3$);

$^{31}$P {$^1$H} NMR (146 MHz, MeOD): δ/ppm 27.31 (d, $^1J_{Rh-P}$ = 160.4 Hz)

MS(ESI), positive mode, in MeOH, *m/z* for **4**, [M]$^+$ (C$_{34}$H$_{38}$N$_2$Na$_3$O$_9$PRh), Calculated: 917.0220, Found: 917.0220.

### 3.2.4. Synthesis of [Rh(bmim)(cod)(pta)]BF$_4$ (**5**)

Method A: In a Schlenk tube under argon atmosphere in 50 mL distilled CH$_2$Cl$_2$, 0.200 g (0.44 mmol) [Rh(OH)(cod)]$_2$ and 0.199 g (0.88 mmol) [bmimH]BF$_4$ were dissolved. The solution was stirred at reflux temperature for 4 h, followed by filtering under argon. A portion of 0.137 g (0.88 mmol) pta was added and stirred for another 2 h at reflux temperature. Then it was filtered again and the solvent was removed by evaporation under vacuum. The sticky residue was cooled in liquid N$_2$ and triturated with small portions (5 mL) of cold diethyl ether yielding 0.332 g (0.559 mmol), 64%, product.

Method B: In a Schlenk tube, 0.050 g (0.129 mmol) [RhCl(bmim)(cod)] (**3**) was dissolved in 4 mL dry MeOH, and 0.014 g (0.129 mmol) NaBF$_4$ was added and stirred at room temperature for 3 h. The NaCl was filtered under argon and 0.020 g (0.13 mmol) pta was added and the mixture was stirred for 3 h at room temperature. Then it was filtered again and the solvent was removed by evaporation under vacuum, the sticky residue was triturated with small portions (5 mL) of cold diethyl ether yielding 0.063 g (0.106 mmol), 82%, product.

$^1$H NMR (360 MHz, MeOD) δ/ppm: 7.37, 7.32 (2H, -N-C*H*=C*H*-N-), 4.84–4.80 (m, 2H, -C*H$_2$*-, cod), 4.52–4.44 (m, 2H, -C*H$_2$*-, cod + 6H, N-C*H$_2$*-N, pta), 4.17-4.09 (m, 2H, N-C*H$_2$*CH$_2$CH$_2$CH$_3$), 3.95-3.92 (m, 3H, N-C*H$_3$* + 6H, P-C*H$_2$*-N, pta), 2.41–2.29 (m, 8H. = CH-C*H$_2$*-, cod), 2.03–1.79 (m, 2H, N-CH$_2$C*H$_2$*CH$_2$CH$_3$), 1.57–1.49 (m, 2H, N-CH$_2$CH$_2$C*H$_2$*CH$_3$), 1.06 (t, 3H, N-CH$_2$CH$_2$CH$_2$C*H$_3$*);

$^{13}$C{$^1$H}NMR (100 MHz, MeOD) δ/ppm: 175.71 (dd, $^1J_{Rh-C}$ = 48.5 Hz, $^2J_{C-P}$ = 18.5 Hz. NCN), 124.14, 121.76 (-N-CH=CH-N-), 100.13 (dd, $^1J_{Rh-C}$ = 8.6 Hz, $^3J_{C-P}$ = 9.9 Hz =CH-CH$_2$-, =CH-CH$_2$-, cod), 98.73 (dd, $^1J_{Rh-C}$ = 7.3 Hz, $^3J_{C-P}$ = 9.9 Hz, =CH-CH$_2$-, cod), 89.12 (dd, $^1J_{Rh-C}$ = 13.9 Hz, $^3J_{C-P}$ = 6.6 Hz, -*C*H=CH-, cod), 71.93 (d, $^3J_{C-P}$ = 6.6 Hz, N-*C*H$_2$-N, pta), 50.53(d, $^3J_{C-P}$ = 13,4 Hz, -*C*H$_2$-P, pta), 50.51 (N-*C*H$_2$CH$_2$CH$_2$CH$_3$), 36.95 (*C*H$_3$-N-), 32.72, 31.27 (-*C*H$_2$-, cod), 30.33 (-N-CH$_2$*C*H$_2$CH$_2$CH$_3$), 30.23, 29.29 (-*C*H$_2$-, cod), 20.07 (-N-CH$_2$CH$_2$*C*H$_2$CH$_3$), 12.98 (N-CH$_2$CH$_2$CH$_2$*C*H$_3$);

$^{31}$P{$^1$H} NMR (146 MHz, MeOD) δ/ppm: -54.90 (d, $^1J_{Rh-P}$ = 125.6 Hz);

MS(ESI), positive mode, in MeOH, *m/z* for **5**, [M]$^+$ (C$_{22}$H$_{38}$N$_5$PRh), Calculated: 506.1914, Found: 506.1915.

### 3.2.5. General Procedure Redox Isomerization of Allylic Alcohols

Catalytic isomerization of allylic alcohols was performed in Schlenk tubes, into which $1 \times 10^{-3}$ mol allylic alcohol and $1 \times 10^{-5}$ mol **1**, **3**, **4**, **5**, or $2 \times 10^{-5}$ mol **1** together with $2 \times 10^{-5}$ mol *m*tppms-Na (in 450 μL MeOH) were dissolved in 3 mL deoxygenated water or aqueous phosphate buffer. All reactions were carried out under oxygen-free atmosphere using argon or nitrogen gas. The reaction mixtures were heated (80 °C) in a thermostated bath and stirred for the desired reaction time, then cooled to room temperature. In the case of water-insoluble substrates, the product mixtures were extracted twice with 1 mL of chloroform, the extracts were dried over $MgSO_4$, and the conversions were determined by gas chromatography or by $^1$H NMR ($CDCl_3$). In the case of water-soluble allylic alcohols, the conversions were determined by $^1$H NMR ($D_2O$). The presented yields are averages of 3–5 measurements, with a reproducibility of ±3%.

In the catalyst recycling experiments, the reaction mixtures was cooled down to room temperature (r.t.). The extraction was done under argon with 2 mL of hexane. Traces of hexane were removed from the aqueous phase by stirring under vacuum for 20 min at r.t., and the resulting aqueous solution of the catalyst was used in the next catalytic cycle.

### 4. Conclusions

We have prepared new water-soluble Rh(I)-NHC complexes with a combination of bmim or sulfonated SIMes (sSIMes) as *N*-hetrocyclic carbene, and mono- or trisulfonated triphenylphosphine (*m*tppms-Na, *m*tppts-Na$_3$) or 1,3,5-triaza-7-phosphadamantane (pta) as phosphine ligands. These complexes proved active catalysts for redox isomerization of various alk-1-en-3-ols in aqueous reaction systems. The reactions were selective to the ketone product and proceeded under inert atmosphere with no need of hydrogen. The water-solubility of the catalysts allowed recycling with modest loss of activity. Based on these attributes, the redox isomerization of allylic alcohols catalyzed by the Rh(I)-NHC-phosphine complexes may serve as the basis for useful synthetic procedures.

**Supplementary Materials:** The following are available online at http://www.mdpi.com/2073-4344/10/11/1361/s1, Table S1. Experimental details of X-ray diffraction measurement; Table S2. Bond lengths for [Rh(bmim)(cod)(pta)]BF$_4$ (**5**); Table S3. Bond angles for [Rh(bmim)(cod)(pta)]BF$_4$ (**5**); Figure S1. GC conditions for analysis of reaction mixtures; Figure S2-S11. $^1$H, $^{31}$P{$^1$H} NMR and $^{13}$C{$^1$H} NMR spectra of [RhCl(cod)(sSIMes)] (**1**), [RhCl(bmim)(cod)] (**3**), Na$_2$[Rh(bmim)(cod)(*m*tppts)] (**4**), [Rh(bmim)(cod)(pta)]BF$_4$ (**5**); Figure S12-15. HR ESI-MS spectra of complexes **1**, **3**, **4**, **5**; Figure S16–18. ORTEP view of [Rh(bmim)(cod)(pta)]BF$_4$ (**5**); Figure S19. packing diagram of [Rh(bmim)(cod)(pta)]BF$_4$ (**5**).

**Author Contributions:** Conceptualization, C.E.C., M.F., F.J.; Methodology, M.F., G.P., A.U., H.H.; Synthesis and characterization of catalysts, M.F., C.E.C., E.T.; Catalysis experiments, C.E.C., M.F., E.T., Discussion of experimental results, all authors; Writing–Original Draft Preparation, all authors; Writing–Review and Editing, C.E.C., F.J. and U.A. All authors have read and agreed to the published version of the manuscript.

**Funding:** The research was supported by the EU and co-financed by the European Regional Development Fund (under the projects GINOP-2.3.2-15-2016-00008 and GINOP-2.3.3-15-2016-00004), and by the Thematic Excellence Programme of the Ministry for Innovation and Technology of Hungary (ED_18-1-2019-0028), within the framework of the Vehicle Industry thematic programme of the University of Debrecen. The financial support of the Hungarian National Research, Development and Innovation Office (FK-128333) is greatly acknowledged.

**Acknowledgments:** The authors are grateful to Attila C. Bényei for recording the X-ray diffraction data. Thanks are also expressed to Cynthia Nagy for the HR ESI-MS measurements and to Tibor Csupász for his efforts in attempted isolation of **2** by preparative HPLC. We are also grateful to Ágnes Kathó for her help and useful advices in preparation of the manuscript.

**Conflicts of Interest:** The authors declare no conflict of interest.

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
