# Peer review of "Redox Isomerization of Allylic Alcohols Catalyzed by New Water-Soluble Rh(I)-N-Heterocyclic Carbene Complexes"

_catalysts, doi:10.3390/catal10111361_

Round 1

Reviewer 1 Report

This study is focused on synthesis of water-soluble, N-heterocyclic carbene (NHC) or mixed N-heterocyclic carbene/tertiary phosphine complexes, analyzed by X-ray diffraction and 1H, 31P{1H} NMR and 13C{1H} NMR, and their application for redox isomerization of allylic alcohols in aqueous media.

This work seems to be of high scientific quality, and the concepts, methods, and results are described/discussed clearly. In my opinion, the manuscript can be accepted in the current format, except very few minor issues/spell checking is required:

Firstly, it would be useful to name the analytical methods that's been utilized for analyzing the synthesized samples, in the last paragraph of the Introduction.

Secondly, please check and the below issues and correct the text accordingly:

line 47: pta should be defined here as it's the first time that it's appeared in the manuscript; instead, the authors defined it in the Conclusion (line448).

line 128: "Another difference is in that the..." Delete "in"

line 217: "Figure 2 shows the time course of the redox isomerization of oct-1-en-3-ol catalyzed by 1 in with..." Delete "in"

line 243: "that the detailed study of the reaction kinetics was not considered part of the present study..." Correct to "as a part"

line 258: "One important reason for application of water-soluble catalysts in organic synthesis is in the..."

Delete "in"

line 267: "After of each catalytic cycle..."

Delete "of"

line 267-268: "...the reaction mixture was cooled to room temperature was extracted with hexane."

What was extracted with hexane? Correct the sentence.

line 285: "In aqueous organometallic catalysis, one of the most important reaction parameter is the..."

Correct to "parameters"

line 286: "... especially in cases when the reactions proceed with formation or consumption H+..."

Correct to "of H+"

line 407: "Than it was filtered again..."

Correct to "Then"

Author Response

Catalysts-1012541

Czégéni et al: “Redox isomerization of allylic alcohols catalyzed by new water-soluble Rh(I)-N-heterocyclic carbene complexes

Response to Reviewer 1

We are grateful to Reviewer 1 for the careful review of our manuscript in a short time period (despite of the difficult global circumstances), and appreciate the favorable comments and helpful suggestions. We agreed with all the suggestions and the text is corrected accordingly; this revision made the text substantially better.

Please, find a list of changes made in the manuscript.  (Line numbers are those of the Revised manuscript, line numbers in parentheses are those in the original submission).

Comments (Q) and responses (R):

C1:      Firstly, it would be useful to name the analytical methods that's been utilized for analyzing the synthesized samples, in the last paragraph of the Introduction

R1:      line 93-94 The analytical techniques used for sample characterization are listed at the end of the Introduction.

C2:       line 47 (47): pta should be defined here as it's the first time that it's appeared in the manuscript; instead, the authors defined it in the Conclusion (line448).

R2:      In the revised manuscript, pta is defined in lines 47.

C3:      line 136 (128): "Another difference is in that the..." Delete "in"

R3:      Deleted.

C4:      line 226 (217): "Figure 2 shows the time course of the redox isomerization of oct-1-en-3-ol catalyzed by 1 in with..." Delete "in

R4:      Deleted.

C5:      line 250 (243): "that the detailed study of the reaction kinetics was not considered part of the present study..." Correct to "as a part"

R5:      Corrected.

C6:      line 265 (258): "One important reason for application of water-soluble catalysts in organic synthesis is in the..."

R6:      Corrected.

C7:      Delete "in" line 274 (267): "After of each catalytic cycle..."

R7:      Deleted.

C8:      Delete "of"  line (267-268): "...the reaction mixture was cooled to room temperature was extracted with hexane." What was extracted with hexane? Correct the sentence.

R8:      line 274-275 The sentence now reads: "...the reaction mixture was cooled to room temperature and it was extracted with hexane."

C9:      line 293 (285): "In aqueous organometallic catalysis, one of the most important reaction parameter is the..."  Correct to "parameters"

R9:      Corrected.

C10:    line 294 (286): "... especially in cases when the reactions proceed with formation or consumption H+..."  Correct to "of H+"

R10:    Corrected.

C11:    line 415 (407): "Than it was filtered again..."  Correct to "Then"

R11:    Corrected.

Reviewer 2 Report

The manuscript by Czégéni et al. reports on the synthesis of new, water-soluble Rh-NHC catalysts, their characterization, and subsequent use in the redox isomerization of various alk-1-en-3-ols in aqueous solution. The use of water as a reaction medium for catalysis is of high interest, especially in terms of facile product isolation (extraction) and at the same time reusability of the catalyst in the aqueous phase. The reported data seem consistent and scientifically sound. The scope of the research fits additionally to the scope of the journal. Therefore, the referee recommends publication of this work in Catalysts after the following issues have been addressed: 

1) The abstract is very difficult to read. The authors may consider rephrasing/removing the extensive abbreviations/nomenclatures and the numbers of the molecules. Additionally, please change "could be reused" to "were reused" in the abstract.

2) Line 49 following: Talking about water-solubility of Rh-NHC catalysts, the authors might want to mention the possibility to incorporate such complexes in a protein (see Klein-Gebbink and co-workers, ChemCommun, 2015, 51, 6792 and Okuda and coworkers, CatalSciTech, 2019, 4, 942). In the referee's personal view, this is one of the best ways to make metal catalysts water-soluble.

3) Please check the consistent use of mol% vs. mol %

4) Please check stöchiometry of the reactions: e.g., in scheme 2 "-HCl" is missing.

5) Synthesis of compound 1 and catalysis: The synthesis of purified 1 failed, however, catalytic runs are reported. The referee would appreciate an elemental analysis of the bulk-product that was used for catalysis to allow better reproducibility of the results.

6) Scheme 4: Is the charge at the Rh of complex 4 correct? The Ligand makes the overall charge neutral and the anion would be missing in that case. Same fpr Scheme 3, compound 2. Additionally, in Scheme 4, "-NaCl" is missing. For the sake of consistency, please check stöchiometry throughout the manuscript.

7) Under your schemes/tables, please give molar concentrations instead of mol for the corresponding compounds.

Author Response

Catalysts-1012541

Czégéni et al: “Redox isomerization of allylic alcohols catalyzed by new water-soluble Rh(I)-N-heterocyclic carbene complexes

Response to Reviewer 2

We are grateful to Reviewer 2 for the questions and positive comments on our manuscript in a short time period (despite of the difficult global circumstances). The manuscript was modified accordingly; our point by point responses are found below. We feel that this revision resulted in a significantly better manuscript.

(Line numbers are those of the Revised manuscript, line numbers in parentheses are those in the original submission).

Comments (C) and responses (R):

C1:      The abstract is very difficult to read. The authors may consider rephrasing/removing the extensive abbreviations/nomenclatures and the numbers of the molecules. Additionally, please change "could be reused" to "were reused" in the abstract.

R1:      The entire abstract has been revised according the Reviewer’s advice. Abbreviations/nomenclatures/numbering have been removed, and the abbreviations are defined at their earliest appearance in the text.

C2:      Line 69 (49) following: Talking about water-solubility of Rh-NHC catalysts, the authors might want to mention the possibility to incorporate such complexes in a protein (see Klein-Gebbink and co-workers, ChemCommun, 2015, 51, 6792 and Okuda and coworkers, CatalSciTech, 2019, 4, 942). In the referee's personal view, this is one of the best ways to make metal catalysts water-soluble.

R2:      Many thanks for this suggestion. The references were modified according to the suggestion of Reviewer 2. The new references can be found in the text at lines 69.
In addition, two related references were added to line 55.

C3:      Please check the consistent use of mol% vs. mol %

R3:      The entire text was checked and the required changes were made throughout.

C4:      Please check stöchiometry of the reactions: e.g., in scheme 2 "-HCl" is missing.

R4:      Corrected.  Similar changes were made in Schemes 3 and 4, too.

C5:      Synthesis of compound 1 and catalysis: The synthesis of purified 1 failed, however, catalytic runs are reported. The referee would appreciate an elemental analysis of the bulk-product that was used for catalysis to allow better reproducibility of the results.

R5:      As it is described in the manuscript, we could not isolate compound 1 sufficiently uniform with clean NMR and ESI mass spectra. These samples were not submitted to elemental analysis. On the other hand, aliquots of a solution of compound 1 (prepared in reaction of [RhCl(cod)]2 and sulfonated SIMes in MeOH in the presence of K2CO3; Scheme 2 in the manuscript) were used for catalysis (after removal of the excess base and KCl) with consistent results: the conversions obtained were within the ± 3 % reproducibility range observed with the other catalysts (e.g. with 5). This is mentioned in the manuscript at lines 120-121 and described in more detail in lines 359-365. No change was made to the manuscript.

C6:      Scheme 4: Is the charge at the Rh of complex 4 correct? The Ligand makes the overall charge neutral and the anion would be missing in that case. Same for Scheme 3, compound 2. Additionally, in Scheme 4, "-NaCl" is missing. For the sake of consistency, please check stöchiometry throughout the manuscript.

R6:        The Reviewer 2 is right; the charge of complex 4 was incorrect in Scheme 4 as well as for compound 2 in Scheme 3. The original Scheme 4 and Scheme 3 have been replaced in the manuscript with redrawn schemes which show the correct charges.

C7:      Under your schemes/tables, please give molar concentrations instead of mol for the corresponding compounds.

R7:      The amounts of the reagents (mol) were replaced with molar concentrations (M) in all cases where catalytic results were reported (Tables 1-3, Figures 2-5).
